# Influences of *Weizmannia coagulans* PR06 Fermentation on Texture, Cooking Quality and Starch Digestibility of Oolong Tea-Fortified Rice Noodles

**DOI:** 10.3390/foods13172673

**Published:** 2024-08-24

**Authors:** Juqing Huang, Pufu Lai, Lihui Xiang, Bin Lin, Weibin Li, Wenquan Yu, Qi Wang

**Affiliations:** 1Institute of Food Science and Technology, Fujian Academy of Agricultural Sciences, Fuzhou 350003, China; jq_huang@zju.edu.cn (J.H.); laipufu@163.com (P.L.); linbin591@163.com (B.L.); liweibin1998@126.com (W.L.); 2Key Laboratory of Processing of Subtropical Characteristic Fruits, Vegetables and Edible Fungi, Ministry of Agriculture and Rural Affairs of China, Fuzhou 350003, China; 3Fujian Key Laboratory of Agricultural Product (Food) Processing, Fuzhou 350003, China; 4Tea Research Institute, Fujian Academy of Agricultural Sciences, Fuzhou 350013, China; xianglihui@faas.cn

**Keywords:** Oolong tea-fortified rice noodles, fermentation by *Weizmannia coagulans* PR06, texture, cooking quality, in vitro starch digestibility

## Abstract

*Weizmannia coagulans* is increasingly employed in food processing owing to its health benefits. Our previous research developed Oolong tea-fortified rice noodles with unique flavor and potent antioxidant activity; however, their texture still requires improvement. In this study, Oolong tea-fortified rice noodles were fermented using *W. coagulans* PR06 at inoculation amounts of 1%, 3%, and 5% (*v*/*v*), and assessed for cooking quality, texture, and starch digestibility. The results indicated that fermentation with 3% and 5% *W. coagulans* PR06 altered the amylopectin length distribution in the rice noodles and increased the degree of starch short-range order. Furthermore, the fermentation process increased the storage modulus (G′) and loss modulus (G″) values, decreased the tan δ value, and strengthened the interactions among tea polyphenols, proteins, and starch in the rice flour gel. Consequently, this process increased the hardness and chewiness of the rice noodles, decreased their broken strip rate and cooking loss, and significantly reduced their in vitro starch digestibility. Overall, fermentation with *W. coagulans* PR06 markedly improved the texture and cooking quality of Oolong tea-fortified rice noodles while effectively delaying starch digestion. This study highlights the potential application of *W. coagulans* PR06 in developing diverse and functional rice noodle products.

## 1. Introduction

Rice noodles, a staple food in many Asian countries, are primarily made from rice flour and water through a series of processes including soaking, grinding, filtering, steaming, and drying [1]. They are highly valued not only for their distinctive texture and flavor but also for their adaptability in a wide range of culinary uses. Tea, a widely consumed beverage, has rich bioactive compounds, including polyphenols, theaflavins, polysaccharides, caffeine and theanine [2]. These compounds are known for their antioxidant, anti-inflammatory, and antimicrobial properties [3,4,5], making tea a valuable ingredient in the food processing industry. The current state of tea-based food processing has seen significant advancements, innovations ranging from tea-infused beverages and snacks to tea-based staple foods, such as Chinese noodles [6,7] and Chinese steamed bread [8]. Adding tea to the noodles and switching from drinking tea to eating tea also has better health effects [7]. Currently, while there are several studies focused on green tea and yellow tea powder noodle products [6,7], research on oolong tea noodle products remains scarce. Oolong tea, a typical semi-fermented tea, is renowned for its elegant aroma and characteristic flavor [9]. Among the most famous varieties is Wuyi rock tea, produced in the Wuyi Mountain and nearby regions, in northern Fujian Province, China [9]. It is highly esteemed by consumers for its distinctive ‘rock charm and floral fragrance’ [9]. Our previous study found that the addition of oolong tea (Wuyi rock tea) to rice noodles imparted a unique flavor, increased antioxidant activity, and reduced starch digestibility, thereby enhancing the nutritional and health properties of the rice noodles [10]. However, the incorporation of oolong tea can significantly reduce the hardness and chewiness of rice noodles [10], which affects the quality and consumer acceptability, necessitating further improvement.

According to the raw material pre-treatment method, rice noodles can be divided into fermented and unfermented. As interest in functional foods continues to grow, fermented rice noodles hold significant potential for both traditional markets and modern consumers seeking health-promoting food options [1]. The fermentation process involves the enzymatic breakdown of rice starches by microorganisms such as lactic acid bacteria, yeasts, and molds. Through the action of microorganisms, fermented rice noodles undergo transformation, resulting in improved texture, chewiness, nutritional value, and distinctive taste [11,12]. However, current research on the starter of fermented rice noodles primarily focus on *Lactobacillus* [11,12], and there is a notable absence of studies on *Weizmannia coagulans*-fermented rice noodles, highlighting the need for further exploration and research.

*Weizmannia coagulans* (formerly *Bacillus coagulans*) is a facultatively anaerobic to microaerobic, Gram-positive, rod-shaped, endospore-forming, and lactic acid-producing bacterium [13]. Several strains of *W. coagulans* have been demonstrated to potentially modulate microbiota composition, enhance immunity, and alleviate metabolic disorders [14,15]. Furthermore, *W. coagulans* produces heat-resistant spores, allowing it to survive processing and cooking, which is particularly advantageous for the food industry [16]. Currently, the application of *W. coagulans* is rapidly expanding across various food sectors [17]. It is being used in the production of fermented dairy products, beverages, and functional foods [17]. Despite its demonstrated benefits in other food products, its application in rice noodle processing remains largely unexplored.

Therefore, this research aims to investigate the influence of *W. coagulans* PR06 fermentation on the texture characteristics, cooking quality and starch digestibility of Oolong tea-fortified rice noodles. This study utilized the unique properties and advantages of oolong tea and *W. coagulans*. Their innovative application in rice noodle processing presents significant potential for improving the overall quality and nutritional value of these popular staple foods.

## 2. Materials and Methods

### 2.1. Materials and Reagents

Early Indica rice “Zhengui Ai” (7.57% protein, 1.35% lipid, 73.15% carbohydrate, 22.21% amylose) was supplied by Yangchun Lihu Xinghao Rice Industry Co., Ltd., (Yangchun, China). Oolong tea (Wuyi rock tea) was provided by Fujian Baitashan Tea Industry Co., Ltd., (Fuzhou, China). The strain *W. coagulans* PR06 was derived from healthy human feces and preserved at the China Center for Type Culture Collection (CCTCC, No. M20211589). The α-amylase from porcine pancreatic (PPA) (A3176, enzyme activity 10 U/mg), amyloglucosidase (AG) (A3306, enzyme activity 260 U/mg) and isoamylase (08124-5MU) were obtained from Sigma Aldrich (St. Louis, MO, USA). The Glucose oxidation kit (GOPOD) was provided by Suzhou Grace Biotechnology Co., Ltd., (Suzhou, China). Fluorescein Isothiocyanate (FITC) was purchased from Shanghai Universal Biotech Co., Ltd., (Shanghai, China). The Rhodamine B was purchased from Shanghai Macklin Biochemical Technology Co., Ltd., (Shanghai, China). All other chemical reagents used were of analytical grade.

### 2.2. Preparation of Fermented Rice Noodles

The rice grains were ground and passed through an 80-mesh sieve to produce rice flour. The rice flour was thoroughly mixed with corn starch and potato starch in a ratio of 16:3:1. The mixture was sterilized under ultraviolet light for 24 h. Next, the flour mixture was blended with Wuyi rock tea soup at 134% of the total weight of the rice flour and starch mixture. The mixture was stirred to ensure uniform dispersion of the rice flour slurries. To prepare the Wuyi rock tea soup, 40 g of rock tea powder was mixed with 1340 g of distilled water. This blender was heated in boiling water for 3 min and then allowed to cool. The resulting filtrate, obtained by passing through a 100-mesh stainless steel sieve, was adjusted to a total weight of 1340 g with distilled water. The strain *W. coagulans* PR06 was incubated in MRS medium at 37 °C for 24 h and then passaged 2 to 3 times. The suspension was centrifuged at 6000× *g* for 10 min at 4 °C and washed twice with phosphate-buffered saline (PBS) at pH 7.4 to obtain a suspension with a concentration of 2.0 × 10^9^ CFU/mL. Subsequently, starter cultures at inoculation amounts (IA) of 1%, 3%, and 5% were added to the uniformly dispersed rice flour slurries. The slurries were fermented at 45 °C without stirring for 10 h and then poured into an HS-60 self-cooked rice noodle processing machine (Shandong Qufu Hongsheng Machinery Co., Ltd., Qufu, China) to produce fresh rice noodles. The fresh rice noodles were aged at 4 °C for 5 h and then dried at 40 °C until the final product’s moisture content was 13%.

### 2.3. Microstructure Observation

#### 2.3.1. Scanning Electron Microscope (SEM)

The rice noodles were cooked for the ideal duration, frozen at −80 °C for 12 h, and dried in a FD-1D-50 freeze dryer (Shanghai Lanyi Industrial Co., Ltd., Shanghai, China) for 48 h. After freeze-drying, the rice noodles were sliced into 2 mm-long segments, coated with gold, and examined with a Gemini SEM 300 scanning electron microscope (Carl Zeiss AG, Jena, Germany). Images of the samples’ cross-sections were captured at an acceleration voltage of 3 kV and magnified 50× and 500×.

#### 2.3.2. Fluorescence Microscope

The distribution of proteins and starch in the rice noodles was determined using an Eclipse Ts2 fluorescence microscope (Nikon Corp, Tokyo, Japan), with some modifications based on previous studies [18]. The rice noodles were sliced into 1.0 mm pieces and subjected to staining with 1 mL of 0.025% fluorescein isothiocyanate (FITC) for 20 min in the dark to visualize starch. Subsequently, they were stained with 1 mL of 0.025% Rhodamine B dye for another 20 min in the dark to detect proteins. The excitation wavelengths for Rhodamine B and FITC were 543 nm and 488 nm, and the emission wavelengths were 570~630 nm and 500~530 nm, respectively.

### 2.4. Texture Measurement

The texture profile of rice noodles was measured following the method of Bae et al. [19], with slight adjustments. Briefly, the rice noodles were cooked for the optimum duration and then drained for 5 s before testing. The texture analysis was conducted using a TA-XT Plus texture analyzer (Stable Micro Systems, London, UK). The P/36 R probe was selected. The sample was compressed to 85% deformation with a pre-test speed of 1 mm/s, a test speed of 0.2 mm/s, and a post-test speed of 10 mm/s, with an interval of 5 s. The trigger force was set to 20 g. Each sample was tested six times, and the experiment was performed in triplicate. Parameters such as hardness, springiness, cohesiveness, gumminess, chewiness, and resilience were recorded from the test curves.

### 2.5. Determination of Cooking Quality

The cooking properties of rice noodles, including cooking time, cooking loss, and water absorption capacity, were assessed using a previously established method [20]. The ideal cooking time was determined by the time at which the white core of the noodles disappeared when squeezed between two glass plates. To measure water absorption capacity, a 25 g sample of noodles was boiled in 400 mL of distilled water until it reached the optimal cooking time, then rinsed in cold water and drained for 30 s before being weighed. The water absorption capacity was calculated by measuring the difference in weight before and after cooking. The cooking loss was reported as the ratio of the weight of the residue in the cooking water to the original weight of the noodle sample, with the residue being obtained through evaporation in an air oven at 105 °C.

The broken strip rate of rice noodle samples was determined using the GB/T 23587 method for Vermicelli in the People’s Republic of China (GAQSIQ and SAC, 2009) [21]. In brief, twenty intact rice noodle bars were immersed in 900 mL of boiling distilled water until they reached the optimal cooking time. The noodles were then collected, and the broken strip rate for each sample was calculated using the following formula: Broken strip rate (%) = (number of noodles that broke after cooking/number of noodles in the starting material) × 100%.

### 2.6. Measurement of Starch Characteristics

#### 2.6.1. Chain Length Distribution of Amylopectin

The chain length distribution of amylopectin was analyzed according to a previous report [22], using a ICS5000+ high-performance anion-exchange chromatography (HPAEC) (Thermo Fisher Scientific, Waltham, MA, USA) equipped with a pulsed-amperometric detector (Dionex Corp, Sunnyvale, CA, USA).

Sample Preparation and Extraction: Starch (10 mg) was dissolved in 5 mL of water in a boiling water bath for 60 min. Sodium azide solution (10 μL, 2% *w*/*v*), acetate buffer (50 μL, 0.6 M, pH 4.4), and isoamylase (10 μL, 1400 U) were added to the starch dispersion, and the mixture was incubated in a water bath at 37 °C for 24 h. The hydroxyl groups of the debranched glucans were reduced by treatment with 0.5% (*w*/*v*) sodium borohydride under alkaline conditions for 20 h. The solution (approximately 600 μL) was dried in vacuo at room temperature and then dissolved in 30 μL of 1 M NaOH for 60 min. Subsequently, the solution was diluted with 570 μL of distilled water.

HPAEC Conditions: The sample extracts were then analyzed using HPAEC on a CarboPac PA-200 anion-exchange column (250 × 4.0 mm i.d., 10 μm). An injection volume of 5 μL was utilized. The mobile phase consisted of two components: A (0.2 M NaOH) and B (0.2 M NaOH and 0.2 M NaAc). The column temperature was maintained at 30 °C throughout the analysis, with a flow rate set to 0.4 mL/min. The gradient program was as follows: 90:10 (*v*/*v*) at 0 min, 90:10 (*v*/*v*) at 10 min, 40:60 (*v*/*v*) at 30 min, 40:60 (*v*/*v*) at 50 min, 90:10 (*v*/*v*) at 50.1 min, and 90:10 (*v*/*v*) at 60 min.

Data Analysis and Interpretation: Data were acquired on the ICS5000+ HPAEC (Thermo Fisher Scientific, Waltham, MA, USA) and processed using Chromeleon 7.2 CDS (Thermo Fisher Scientific, Waltham, MA, USA). Amylopectin was classified into four groups based on the degree of polymerization (DP) into four groups: A (DP 6~12), B1 (DP 13~24), B2 (DP 25~36), and B3 (DP ≥ 37). The area ratio (%) of each peak was calculated for these four groups.

#### 2.6.2. Short-Range Ordered Structure of Starch

The short-range ordered structure of starch in rice noodles was determined using Fourier-transform infrared spectroscopy (FTIR), as described in a previous report [20]. The 150-mesh sieved rice noodle powder and anhydrous KBr were mixed at a 1:100 (*w*/*w*) ratio, thoroughly ground multiple times, and then pressed into a transparent pellet using a pellet press. The spectra were recorded over the range of 4000 to 400 cm^−1^ using a Nicolet iN10 FTIR spectrometer (Thermo Fisher Scientific, Waltham, MA, USA). The number of scans was set to 32, with a resolution of 4 cm^−1^. The 1200 to 800 cm^−1^ region of the spectrum was deconvolved using OMNIC™ Specta Software Version 8.0. The ratio of absorbances at 995 cm^−1^/1022 cm^−1^ (R 995/1022) was calculated to estimate the short-range ordered structure of starch [6].

#### 2.6.3. In Vitro Digestibility of Starch 

The starch digestibility of rice noodles was determined using a rapid in vitro assay, as described in a previous report [6]. Approximately 0.5 g of rice noodle powder was mixed with distilled water at a 1:2 weight ratio and boiled for 10 min. After cooling, pepsin (pH 2.0) was added and the mixture was incubated in continuously shaking water bath at 37 °C for half an hour. The pH was then adjusted to 5.2 using NaOH solution and sodium acetate buffer. Subsequently, the activated mixed enzyme solution (comprising PPA and AAG at a 20:1 weight ratio) was added and allowed to react in a shaking water bath at 37 °C for varying intervals (20, 120, and 180 min). The enzyme was then deactivated with anhydrous ethanol. The glucose content at different time points was determined using the GOPOD assay kit. Starch was classified into three categories: rapidly digestible starch (RDS), slowly digestible starch (SDS), and resistant starch (RS). RDS was completely digested within 20 min, SDS was digested between 20 and 120 min, and RS remained undigested after 120 min. The contents of RDS, SDS, and RS were calculated using the following Equations (1)–(3).
(1)$$\mathrm{RDS}(\%)=\frac{{(G}_{20}-G_{0})\times 0.9}{\mathrm{TS}}\times 100$$
(2)$$\mathrm{SDS}(\%)=\frac{{(G}_{120}-G_{20})\times 0.9}{\mathrm{TS}}\times 100$$
(3)$$\mathrm{RS}(\%)=\frac{(\mathrm{TS}-(\mathrm{RDS}+\mathrm{SDS}))}{\mathrm{TS}}\times 100$$
where G_20_ and G_120_ represent the glucose released within 20 min and 120 min, respectively; FG denotes the amount of free glucose prior to enzymatic hydrolysis; and TS stands for total starch.

### 2.7. Measurement of Rheological Properties

The rheological properties of rice flour gel were measured according to a previous study [23] with slight modifications. Five grams of rice noodle powder, which had passed through an 80-mesh sieve, were mixed with 25 mL of distilled water. After stirring and soaking for 20 min, the mixture was placed in a MCR-102e Anton Paar rheometer (Anton Paar, Graz, Austria) to evaluate its rheological properties.

#### 2.7.1. Determination of the Viscosity-Temperature Curve

The viscosity-temperature curve was measured using a coaxial cylinder platform. The strain was set to 1%, and the temperature program was as follows: heating from 25 °C to 95 °C at a rate of 5 °C/min, then cooling from 95 °C to 25 °C at a rate of 2 °C/min. The effects of varying temperatures on the apparent viscosity of both unfermented and fermented samples were investigated.

#### 2.7.2. Dynamic Frequency Sweep Tests

A frequency sweep was conducted using a coaxial cylinder platform. The test temperature was maintained at 25 °C, with the strain set to 1%. Changes in the sample’s the storage modulus (G′), loss modulus (G″), and loss angle (tan δ = G″/G′) over an angular frequency range of 0.1 to 100 rad/s were measured.

### 2.8. Analysis of Thermal Properties

The thermal properties of rice noodles were determined using a PerkinElmer DSC Diamond-I with an internal coolant, as described in a previous study [24]. Nitrogen was used as purge gas. Each sample, with a moisture content of 70% (*w*/*v*), was placed in a high-pressure stainless steel pan and sealed with a gold-plated copper seal. The samples were then scanned from 30 °C to 140 °C at a heating rate of 10 °C/min. The onset temperature (To), conclusion temperature (Tc), gelatinization temperature range (To–Tc), and enthalpy (∆H) of gelatinization were recorded from the DSC endothermic curve.

### 2.9. Determination of Antioxidant Capacity

#### 2.9.1. Total Polyphenolic Content (TPC)

The TPC of rice noodles was determined by the Folin–Ciocalteu colorimetric method, according to ISO 14502-1 (2005) [25]. The results were expressed as mg GAE (gallic acid equivalents) per gram of dry weight (DW).

#### 2.9.2. Antioxidant Capacity

The FRAP method was used to determine the total antioxidant capacity of rice noodles, following the instructions of the assay kit (Beyotime Biotechnology Co., Ltd., Shanghai, China).

### 2.10. Statistical Analysis

All data are presented as mean ± standard deviation (SD) (n = 3). The experimental data were analyzed using one-way analysis of variance (ANOVA) followed by Duncan’s test, conducted with IBM SPSS Statistics 25 software. A *p*-value of less than 0.05 was considered statistically significant.

## 3. Results and Discussion

### 3.1. Appearance and Microstructure of Rice Noodles

Figure 1 presents the morphology and SEM micrographs of both non-fermented and fermented Oolong tea-fortified rice noodles. All samples exhibited a golden color similar to that of soup (Figure 1(A1–D1)). The fermented samples appeared slightly darker than the control (Figure 1(A1–D1)). The SEM micrographs reveal that the control rice noodles (Figure 1(A2,A3)) exhibited larger pores and a sparse structure, whereas the rice noodles fermented with *W. coagulans* PR06 (Figure 1(B2–D3)) displayed a more continuous and dense matrix.

The interactions among tea polyphenols, proteins and starch in the system was demonstrated using single fluorescence labeling images (Figure 2(A1–D1)) and multichannel superposition images (Figure 2(A2–D2)) under a fluorescence microscope. In the rice noodle samples, starch was stained with FITC and proteins with Rhodamine B. Consequently, starch appears green, protein appears red, and tea polyphenols naturally exhibit green fluorescence due to their intrinsic self-fluorescence characteristics [6]. The yellow regions indicate the matrix formed by their binding [6,26]. In all samples, the starch and tea polyphenols exhibited specific fluorescence intensity, consistent with previous studies [6,18,27]. Interestingly, the yellow fluorescence area in the samples gradually increased with the increment of IA. This result suggests that *W. coagulans* PR06 fermentation could promote the tea polyphenols-protein-starch interactions in rice noodles, leading to the formation of a more stable structure.

### 3.2. Texture Profile of Rice Noodles

The texture characteristics after cooking are crucial factors influencing the quality and consumer acceptability of rice noodles. Table 1 summarizes the texture features of both non-fermented and fermented Oolong tea-fortified rice noodles. Generally, all texture feature values of Oolong tea-fortified rice noodles increased after fermentation. An increase in IA tended to enhance hardness, springiness, resilience, and cohesiveness, resulting in a gradual rise in gumminess (i.e., hardness × cohesiveness) and chewiness (i.e., hardness × cohesiveness × springiness). Notably, Oolong tea-fortified rice noodles fermented with 3% and 5% *W. coagulans* PR06 exhibited higher springiness and chewiness compared to the control and samples fermented with 1% *W. coagulans* PR06 (*p* < 0.05). This indicates that *W. coagulans* PR06 fermentation can improve the texture of Oolong tea-fortified rice noodles, resulting in a chewier texture. These results are consistent with previous studies [11,12], where fermentation led to increased hardness and chewiness of rice noodles. The improvement in texture casused by *W. coagulans* PR06 fermentation was likely due to changes in the starch composition of rice noodles. The microorganism consumed short-chain amylopectin during fermentation. Consequently, the gel network of fermented rice noodles became denser with enhanced gel strength, resulting in greater hardness, elasticity, and chewiness. Park et al. found that the increased gel hardness was due to the lower pH caused by fermentation [28]. In this study, *W. coagulans* PR06 fermentation also lowered the pH of rice noodles (the pH values of the non-fermented, 1%, 3%, and 5% *W. coagulans* PR06 fermented rice noodles were 6.4, 5.2, 4.4, and 4.2, respectively). These factors, combined with changes in tea polyphenols-protein-starch interactions caused by fermentation (Figure 2), could increase the gel hardness of fermented Oolong tea-fortified rice noodles.

### 3.3. Cooking Quality of Rice Noodles

The primary indicators for evaluating the cooking quality of rice noodles include the broken strip rate, cooking loss, water absorption, and cooking time [19]. The values of these indicators of non-fermented and fermented oolong tea-fortified noodles are listed in Table 2. It was observed that *W. coagulans* PR06 fermentation significantly decreased the broken strip rate, cooking loss and water absorption, while increasing the cooking time of Oolong tea-fortified rice noodles (*p* < 0.05). The results indicated that *W. coagulans* PR06 fermentation altered the structure of Oolong tea-fortified rice noodles. We speculated that rice noodles fermented with *W. coagulans* PR06 developed a more stable and denser network structure through interactions among tea polyphenols, protein, and starch (Figure 2), thereby reducing the broken strip rate while cooking loss and increasing the cooking time. The reduction in water absorption might be attributed to the fermentation of *W. coagulans* PR06, which facilitated the reaction between hydroxyl (OH) groups in tea polyphenols and starch molecules, thereby inhibiting starch’s water absorption.

### 3.4. Starch Characteristics of Rice Noodles

#### 3.4.1. Chain Length Distribution of Amylopectin

Amylopectin has been reported to be digested more vigorously than amylose by starch-utilizing microorganisms during fermentation [29], due to the greater vulnerability of the amorphous region of amylopectin compared to ordered crystallites [23]. The hydrolysis of amylopectin is also evident from the modified chain length distribution profile in the fermented sample (Figure 3 and Table 3). In the fermented sample (3% IA or 5% IA), amylopectin exhibited a significantly lower proportion of A chains (DP 6~12) and a significantly higher proportion of B1 chains compared to the non-fermented sample (*p* < 0.05), indicating the loss of short chains after fermentation. Studies have shown that when amylopectin contains more A chains, it cannot form a stable double helix structure due to a high degree of intermolecular looseness, resulting in a low degree of molecular chain rearrangement and starch regeneration [30]. Additionally, longer molecular chains generally increase steric hindrance, while a high proportion of B1 chains (DP 13~24) facilitates regeneration [31]. During rice noodle preparation, the starch swells and gelatinizes to form a gelatinized state. After a short period, it spontaneously re-agglomerates to form insoluble starch, a process known as retrogradation or starch aging [32]. Retrogradation partially crystallizes the starch within the rice noodles, enhancing their chewiness during consumption [33]. In this study, *W. coagulans* PR06 fermentation increased the proportion of B1 chains while decreasing the proportion of A chains in the amylopectin of Oolong tea-fortified rice noodles. This favored starch regeneration or aging, which partly contributed to the lower broken strip rate and the chewier texture of the fermented samples (Table 1 and Table 2).

#### 3.4.2. Short-Range Ordered Structure of Starch

The FTIR spectra of Oolong tea-fortified rice noodles with/without *W. coagulans* PR06 fermentation are shown in Figure 4. All samples exhibited absorption peaks at 3431, 2926, 1463, 1162, 1083, 932, and 763 cm^−1^, characteristic of starch polysaccharides [34]. Among these peaks, the broad peak at 3431 cm^−1^ corresponds to the vibration of the O-H, 2926 cm^−1^ corresponds to C-H stretching region, 1463 cm^−1^ belongs to C-H stretching region, 1163 and 1083 cm^−1^ belongs to the the vibration of C-O-C of glucose, 932 cm^−1^ is considered the characteristic peak of the absorption peak of the α-1,4 glycosiside bond; 763 cm^−1^ is the vibration of the C-C bond.

The FTIR spectrum bands of starch ranging from 1200 cm^−1^ to 800 cm^−1^ are particularly sensitive to the starch molecular conformation, especially short-range ordered structures, due to the sensitivity of these bands to C-C and C-O stretching and C-H-O bending modes [35]. The 1200 cm^−1^ to 800 cm^−1^ region of the original FTIR spectrum was deconvolved to produce Figure 4B. The band at 995 cm^−1^ associated with the single helix crystalline structure, related to hydrogen bonds of anhydroglucose unit in amylose and is sensitive to water [36]. The band at 1022 cm^−1^ is sensitive to the amorphous structure of rice gel [37]. The short-range ordered structure in starch refers to the structure formed by orderly accumulation in a short distance between double helices, which can be characterized by the absorbance ratio at 995 cm^−1^/1022 cm^−1^ (R 995/1022). Table 3 shows that the R 995/1022 values of Oolong tea-fortified rice noodles fermented with 3% and 5% *W. coagulans* PR06 were significantly higher than those of the control group (*p* < 0.05). This phenomenon indicates that *W. coagulans* PR06 fermentation may facilitate starch reassembly, thereby enhancing the short-range ordered structures of starch (Table 3). Consistent findings were reported by Lu et al. [38], where microbial fermentation led to the formation of more ordered crystalline starches among branched-chain molecules, resulting in increased short-range ordering.

#### 3.4.3. In Vitro Digestibility of Starch

Starch digestibility significantly affects its nutritional function [39]. Starch can be divided into RDS, SDS, and RS [40]. The starch fractions of non-fermented and fermented Oolong tea-fortified rice noodles are shown in Table 3. It was found that 5% *W. coagulans* PR06 fermentation significantly (*p* < 0.05) decreased the content of RDS from 67.81% to 60.53%, while increased the content of RS from 12.46% to 20.71%. During cooking, the rice noodles were notably eroded by water, resulting in a more open starch structure that was increasingly susceptible to digestive enzyme attack [6]. Oolong tea-fortified rice noodles fermented with *W. coagulans* PR06 formed a more stable and denser network structure through tea polyphenols-protein-starch interactions (Figure 2). This increased spatial site resistance, preventing digestive enzymes from contacting the starch, and thus significantly delayed starch digestion (Table 3). In our previous study, adding 4% Oolong tea (Wuyi rock tea) could reduce the proportion of RDS in rice noodles by 9.83%, while increasing the proportion of RS by 6.54% [10]. The results in this study demonstrated that *W. coagulans* PR06 fermentation further decreased the proportion of RDS and increased the proportion of RS, offering an effective method to enhance the nutritional and health qualities of Oolong tea-fortified rice noodles.

### 3.5. Rheological Properties of Rice Noodles

The primary indicators of rheological properties include viscosity, elasticity, and viscoelasticity [23]. The rheological properties of both non-fermented and fermented Oolong tea-fortified rice noodles were analyzed, and the results are shown in Figure 5.

Figure 5A illustrates the apparent viscosity changes with temperature in Oolong tea-fortified rice noodles, both with and without *W. coagulans* PR06 fermentation. It is evident that all *W. coagulans* PR06 fermented samples exhibited significantly higher apparent viscosity compared to the control (*p* < 0.05). Additionally, the apparent viscosity of all samples remained stable during the cooling period. The elasticity and viscoelasticity can be confirmed by the magnitudes of energy stored and lost in the gels under deformation (G′ and G″, respectively) [18]. Generally, G′ is positively correlated with the strength and elasticity of the gels, whereas G″ is positively correlated with their viscosity [18]. As depicted in Figure 5B,C, the storage or elastic modulus (G′) and the loss or viscous modulus (G″) values for all samples exhibited consistent changes. Both the G′ and G″ values of all samples increased with angular frequency (0.1~100 rad/s), and the G′ value was significantly greater than the G″ value for all samples, indicating that the gels were more elastic than viscous, exhibiting characteristics similar to solids. The G′ and G″ values of fermented samples were significantly higher than those of the control (*p* < 0.05), suggesting that fermentation by *W. coagulans* PR06 enhanced both the elasticity and viscosity of the rice flour gel. The rice flour gel forms through a network of swollen starch granules associated with proteins and tea polyphenols [6]. Fermentation by *W. coagulans* PR06 promoted the network interactions among these components (Figure 2), resulting in a more stable structure for the rice flour gel.

The loss tangent (tan δ = G′′/G′) is a more sensitive parameter than G′ or G′′ for detecting changes in the viscoelastic nature of polymer gels [23]. As shown in Figure 5D, the value of tan δ for all samples in the frequency range of 0.1~100 rad/s was less than 1.0, indicating a gel structure that was primarily elastic. The variation profile of tan δ with angular frequency exhibited an opposite trend to G′ and G″ (Figure 5D). Additionally, compared to the non-fermented sample, the tan δ value of the fermented samples was significantly lower (*p* < 0.05), indicating that fermentation had a greater impact on the elasticity than the viscosity of the rice flour gel, making it more solid. It has been reported that well-cross-linked network structures invariably exhibit tan δ values lower than 0.1 [41]. In this study, the tan δ values of all fermented samples were below 0.1, indicating better-formed network structures in comparison to the non-fermented sample. This result aligns with the enhanced network formed by tea polyphenols, proteins, and starch interactions in the fermented samples, as depicted in Figure 2.

### 3.6. Thermodynamic Property of Rice Noodles

Previous studies have demonstrated that the thermal properties of rice flours can be influenced by fermentation [42,43]. Therefore, we measured the thermal profiles of both non-fermented and fermented Oolong tea-fortified rice noodles. As shown in Table 4, the To and Tp of samples in all groups did not exhibit a significant difference (*p* > 0.05). *W. coagulans* PR06 tended to increase the Tc of Oolong tea-fortified rice noodles, but this change was not statistically significant (*p* > 0.05). Compared to the control, the Tc-To of the fermented samples (3% IA and 5% IA) was significantly increased (*p* < 0.05), indicating that the starch in the fermented samples had a more ordered crystalline structure than the non-fermented sample. As shown in Table 3, the ΔH of Oolong tea-fortified rice noodles fermented with 3% and 5% *W. coagulans* PR06 was significantly higher than those of the control group (*p* < 0.05). This result aligned with previous reports [43]. The enthalpy of gelatinization (∆H) refers to the energy needed to disrupt the double helix in starch during gelatinization [38]. The increase in ΔH indicated that more energy was required for the melting of crystalline structures formed by the association between adjacent double helices of starch molecules in rice noodles [44]. In the present study, *W. coagulans* PR06 fermentation increased the proportion of B1 chains while decreasing the proportion of A chains in the amylopectin of Oolong tea-fortified rice noodles (Figure 5 and Table 3). This process promoted the development of an ordered starch structure (Figure 4B and Table 3), subsequently forming a more ordered crystalline structure and resulting in an increase in ΔH.

### 3.7. Antioxidant Capacity of Rice Noodles

Tea polyphenols are known to eliminate oxygen free radicals. Our previous study demonstrated that rice noodles fortified with oolong tea exhibit significantly higher antioxidant capacity compared to the control [10]. Therefore, it was essential to determine the antioxidant capacity of both fermented and non-fermented Oolong tea-fortified rice noodles. The total phenolic content (TPC) and total antioxidant capacity (T-AOC) of Oolong tea-fortified rice noodles, with and without *W. coagulans* PR06 fermentation are compared in Table 4. It can be observed that *W. coagulans* PR06 fermentation did not alter the TPC and T-AOC values of Oolong tea-fortified rice noodles (*p* > 0.05), suggesting that the structure of tea polyphenols was not affected by *W. coagulans* PR06 fermentation.

## 4. Conclusions

The results of this study indicate that the fermentation of *W. coagulans* PR06 (3% and 5% IA) altered the amylopectin length distribution in Oolong tea-fortified rice noodles, as evidenced by an increased proportion of B1 chains (DP 13~24) and a decreased proportion of A chains (DP 6~12). Moreover, the fermentation process increased the degree of short-range order in starch, augmented the G′ and G″ values, decreased the tan δ value, and reinforced the tea polyphenols-protein-starch interactions in the rice flour gel. As a result, this process increased the hardness and chewiness of the rice noodles, while simultaneously decreasing the broken strip rate and cooking loss, and significantly delaying the digestion of starch. Overall, fermentation by *W. coagulans* PR06 significantly enhanced the textural characteristics and cooking quality of Oolong tea-fortified rice noodles, while effectively reducing their starch digestibility. The fermentation by *W. coagulans* PR06 presents a viable method for developing diversified and functional rice noodles with substantial application value.

## Figures and Tables

**Figure 1 foods-13-02673-f001:**
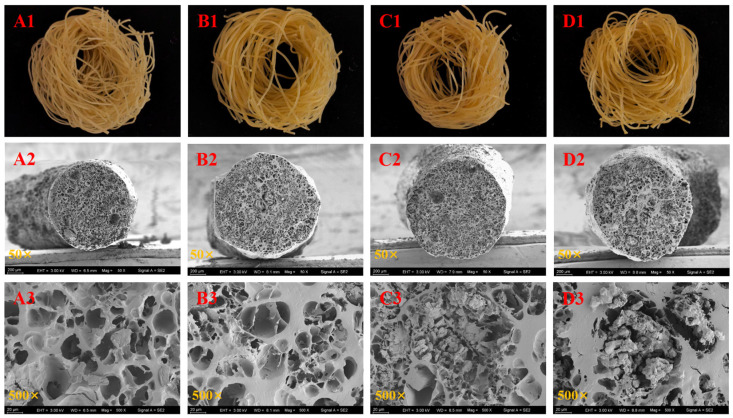
The morphology and SEM micrographs of Oolong tea-fortified rice noodles with/without *W. coagulans* PR06 fermentation. (**A**): Control, (**B**): 1% IA (inoculation amount), (**C**): 3% IA, (**D**): 5% IA. (**A1**–**D1**) represent the morphology of rice noodles while (**A2**–**D3**) show the SEM micrographs of rice noodles.

**Figure 2 foods-13-02673-f002:**
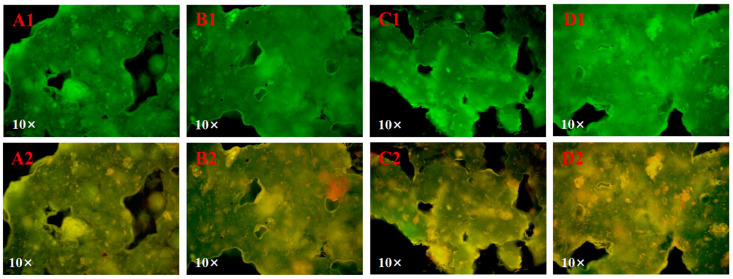
The fluorescence microscope images of Oolong tea-fortified rice noodles with/without *W. coagulans* PR06 fermentation (single fluorescence labeling (**A1**–**D1**) and multichannel superposition (**A2**–**D2**)). (**A**): Control, (**B**): 1% IA (inoculation amount), (**C**): 3% IA, (**D**): 5% IA. Note: Starch appears green while protein appears red, with tea polyphenols naturally appearing green due to its self-fluorescence characteristics. The yellow regions indicated the matrix formed by their binding.

**Figure 3 foods-13-02673-f003:**
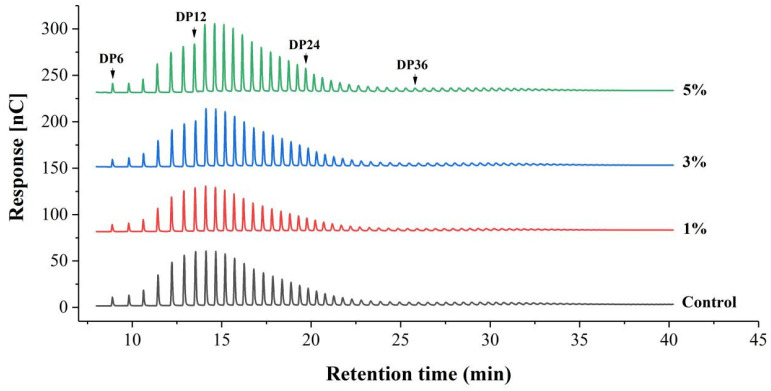
HPAEC elution profiles of amylopectin of Oolong tea-fortified rice noodles with/without *W. coagulans* PR06 fermentation. The control group is non-fermented sample. The 1%, 3%, and 5% groups represent *W. coagulans* PR06 fermented samples with inoculation amounts (IA) of 1%, 3%, and 5%, respectively. Numbers above the peaks indicate the degree of polymerization of maltooligosaccharides.

**Figure 4 foods-13-02673-f004:**
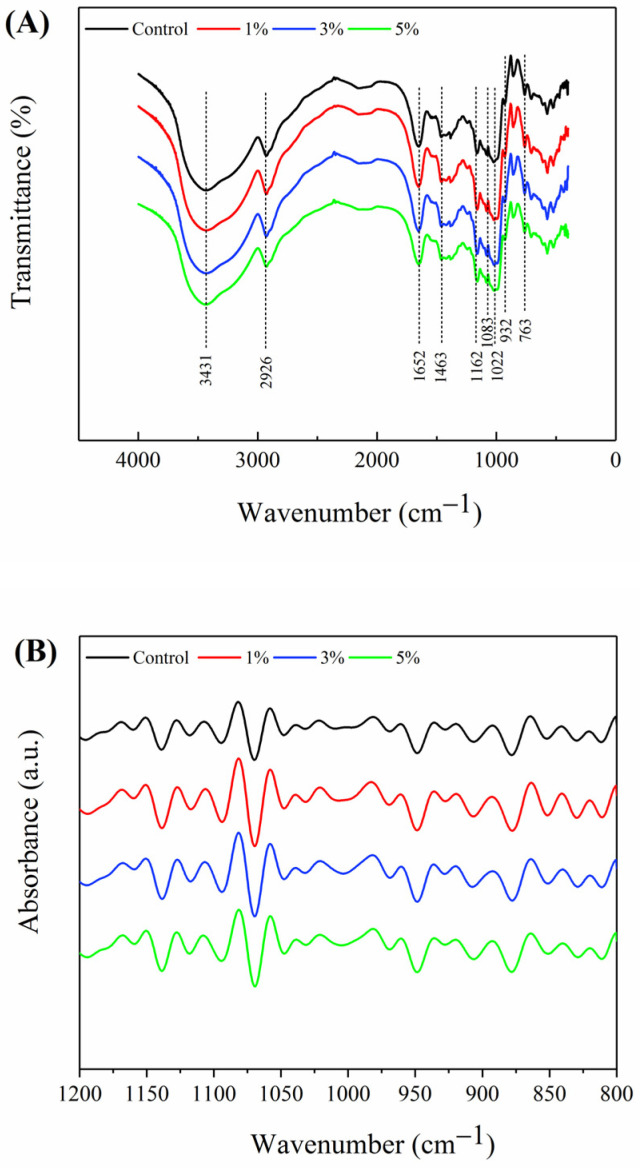
FTIR spectra of Oolong tea-fortified rice noodles with/without *W. coagulans* PR06 fermentation. (**A**) FTIR spectra in the range of 4000–400 cm^−1^, (**B**) FTIR spectra in the range of 1200–800 cm^−1^. The control group is non-fermented sample. The 1%, 3%, and 5% groups represent *W. coagulans* PR06 fermented samples with inoculation amounts (IA) of 1%, 3%, and 5%, respectively.

**Figure 5 foods-13-02673-f005:**
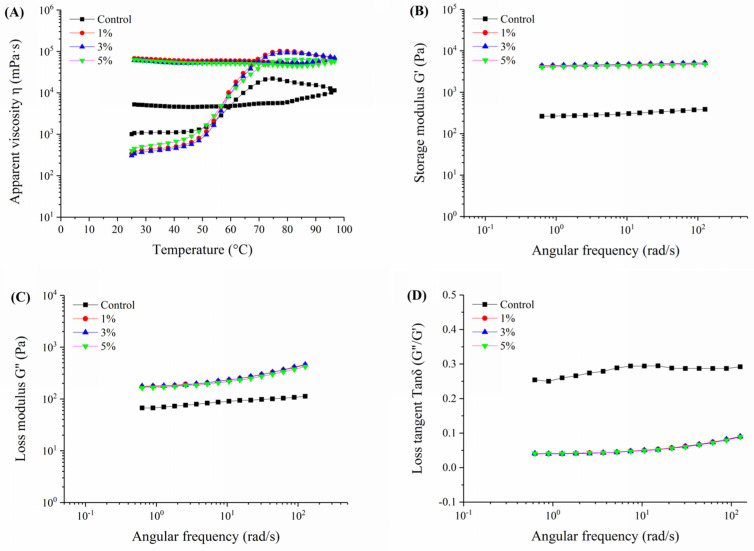
Rheological properties of Oolong tea-fortified rice noodles with/without *W. coagulans* PR06 fermentation. (**A**) change profile of apparent viscosity with temperature (**B**) change profile of storage modulus (G′) with angular frequency, (**C**) change profile of loss modulus (G″) with angular frequency, (**D**) change profile of loss tangent (tan δ) with angular frequency. The control group is non-fermented sample. The 1%, 3%, and 5% groups represent *W. coagulans* PR06 fermented samples with inoculation amounts (IA) of 1%, 3%, and 5%, respectively.

**Table 1 foods-13-02673-t001:** Texture profile of fermented and non-fermented Oolong tea-fortified rice noodles.

Characteristic	Control	*W. coagulans* PR06 Fermented Samples with Different IA		
		1%	3%	5%
Hardness (g)	3468.95 ± 576.28 ^a^	3679.43 ± 454.88 ^a^	4019.51 ± 449.28 ^a^	4264.92 ± 220.53 ^a^
Springiness	0.70 ± 0.06 ^a^	0.81 ± 0.09 ^ab^	0.97 ± 0.09 ^b^	0.91 ± 0.04 ^b^
Cohesiveness	0.83 ± 0.06 ^a^	0.95 ± 0.08 ^a^	0.99 ± 0.05 ^a^	0.94 ± 0.09 ^a^
Gumminess	2914.50 ± 696.11 ^a^	3469.07 ± 413.03 ^a^	3972.56 ± 508.05 ^a^	4007.82 ± 520.39 ^a^
Chewiness	2025.55 ± 455.00 ^a^	2803.27 ± 303.67 ^a^	3798.74 ± 199.48 ^b^	3731.31 ± 546.70 ^b^
Resilience	1.18 ± 0.09 ^a^	1.37 ± 0.09 ^a^	1.35 ± 0.08 ^a^	1.33 ± 0.07 ^a^

IA, Inoculation amount. Different letters in the same row indicated significant differences (*p* < 0.05).

**Table 2 foods-13-02673-t002:** Cooking quality of fermented and non-fermented Oolong tea-fortified rice noodles.

Characteristic	Control	*W. coagulans* PR06 Fermented Samples with Different IA		
		1%	3%	5%
Broken strip rate (%)	5.7 ± 0.4 ^d^	4.5 ± 0.6 ^c^	3.2 ± 0.3 ^b^	2.4 ± 0.4 ^a^
Cooking loss (%)	4.8 ± 0.2 ^b^	4.3 ± 0.2 ^b^	3.3 ± 0.5 ^a^	3.6 ± 0.4 ^a^
Water absorption (%)	191.7 ± 3.1 ^b^	185.7 ± 6.0 ^ab^	179.3 ± 5.1 ^a^	178.7 ± 6.5 ^a^
Cooking time (s)	164.3 ± 4.0 ^a^	187.7 ± 2.1 ^b^	225.3 ± 4.5 ^d^	216.0 ± 5.3 ^c^

IA, Inoculation amount. Different letters in the same row indicated significant differences (*p* < 0.05).

**Table 3 foods-13-02673-t003:** Starch characteristics of fermented and non-fermented Oolong tea-fortified rice noodles.

Characteristic	Control	*W. coagulans* PR06 Fermented Samples with Different IA		
		1%	3%	5%
Chain length distribution of amylopectin (%)				
A (DP 6~12)	24.66 ± 1.33 ^b^	24.79 ± 1.12 ^b^	21.90 ± 1.37 ^a^	20.16 ± 1.80 ^a^
B1 (DP 13~24)	52.44 ± 1.17 ^a^	52.53 ± 2.39 ^a^	55.13 ± 2.12 ^ab^	57.13 ± 1.39 ^b^
B2 (DP 25~36)	12.81 ± 1.18 ^a^	12.91 ± 1.01 ^a^	12.75 ± 0.77 ^a^	12.69 ± 0.80 ^a^
B3 (DP ≥ 37)	10.10 ± 0.38 ^a^	9.78 ± 0.61 ^a^	10.22 ± 1.47 ^a^	10.03 ± 0.84 ^a^
Starch short-range ordered structure				
FTIR ratio (R 995/1022)	0.43 ± 0.05 ^a^	0.47 ± 0.05 ^a^	0.57 ± 0.06 ^b^	0.58 ± 0.10 ^b^
Starch fractions				
RDS (%)	67.82 ± 2.16 ^c^	63.72 ± 1.94 ^bc^	58.49 ± 1.84 ^a^	60.53 ± 2.37 ^ab^
SDS (%)	19.72 ± 1.28 ^ab^	21.19 ± 1.18 ^ab^	22.70 ± 1.87 ^b^	18.76 ± 1.25 ^a^
RS (%)	12.46 ± 1.07 ^a^	15.09 ± 1.44 ^a^	18.81 ± 2.01 ^b^	20.71 ± 1.27 ^b^

IA, Inoculation amount. Different letters in the same row indicated significant differences (*p* < 0.05).

**Table 4 foods-13-02673-t004:** Thermal properties and antioxidant ability of fermented and non-fermented Oolong tea-fortified rice noodles.

Characteristic	Control	*W. coagulans* PR06 Fermented Samples with Different IA		
		1%	3%	5%
To (°C)	46.96 ± 0.32 ^a^	47.34 ± 1.03 ^a^	46.55 ± 0.65 ^a^	46.31 ± 1.05 ^a^
Tp (°C)	55.12 ± 0.53 ^a^	56.04 ± 0.99 ^a^	55.76 ± 1.25 ^a^	56.56 ± 1.01 ^a^
Tc (°C)	64.13 ± 0.92 ^a^	64.81 ± 1.16 ^a^	65.45 ± 1.39 ^a^	65.37 ± 1.56 ^a^
Tc-To (°C)	17.17 ± 0.63 ^a^	17.47 ± 0.56 ^a^	18.91 ± 0.76 ^b^	19.06 ± 0.56 ^b^
ΔH (J/g)	0.78 ± 0.02 ^a^	0.84 ± 0.04 ^a^	1.31 ± 0.04 ^b^	1.34 ± 0.08 ^b^
TPC (mg GAE/g DW)	2.53 ± 0.21 ^a^	2.51 ± 0.19 ^a^	2.59 ± 0.23 ^a^	2.54 ± 0.17 ^a^
T-AOC (mM Fe_2_SO_4_/g)	9.16 ± 0.63 ^a^	9.11 ± 0.57 ^a^	9.33 ± 0.72 ^a^	9.24 ± 0.68 ^a^

IA, Inoculation amount. TPC, total phenolic content. T-AOC, total antioxidant capacity. Different letters in the same row indicated significant differences (*p* < 0.05).

## Data Availability

The original contributions presented in the study are included in the article, further inquiries can be directed to the corresponding author.

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
