# Peer review of "Influences of Weizmannia coagulans PR06 Fermentation on Texture, Cooking Quality and Starch Digestibility of Oolong Tea-Fortified Rice Noodles"

_foods, 2024, doi:10.3390/foods13172673_

Round 1

Reviewer 1 Report

Comments and Suggestions for Authors

-          This paper correspond for scope of journal.

-          The title corresponds to the content of the paper. 

The contribution of study is establishing of oolong tea and Weizmannia coagulans, their innovative application in rice noodle processing presents significant potential for enhancing the overall quality and nutritional value of these popular staple foods.

-          This study addressed to developing technology process for production Oolong tea-fortified rice noodles, with improved  cooking quality, texture, and starch digestibility after fermentation by Weizmannia coagulans PR06 at inoculation amounts of 1%, 3%, and 5% (v/v) process for developing.

-          The aim of research  was to investigate the influence of W. coagulans PR06 fermentation on the texture characteristics, cooking quality and starch digestibility of Oolong tea fortified rice noodles.. The aim is clear pointed out in line from 80 to 82. However, my suggestion is that author write aim as particular last paragraph of chapter Introduction.

-          Key words are appropriately chosen!

-          Scientific methodology is applied correctly. The used method described in detail and useful for researcher in their further studies.

-          Results are clearly presented and discussed.

-          Tables, figures, pictures are clear.

-          The conclusion chapter was written on the base of obtained results.

-          Manuscript is acceptable after minor corrections.

Sugestion:

Sentence from line 82 to 82 :” The findings of this study will provide a scientific basis for the fur ther exploration and development of fermented tea-based rice noodles”- should be delete.

I propose to use only one term Weizmannia coagulans in the text. It is need to state in one sentence that the old name is Bacillus coagulans, but then need use the name used in the title of the paper.

Author Response

We would like to express our gratitude for your professional review, constructive comments, and valuable suggestions regarding our manuscript. We have systematically addressed and revised the manuscript based on your feedback. We look forward to your further review of our updated submission.

Comments 1: Sentence from line 82 to 82 :” The findings of this study will provide a scientific basis for the fur ther exploration and development of fermented tea-based rice noodles”- should be delete.

Response 1: We agree with this comment. Consequently, we deleted the aforementioned sentence.

Comments 2: I propose to use only one term Weizmannia coagulans in the text. It is need to state in one sentence that the old name is Bacillus coagulans, but then need use the name used in the title of the paper.

Response 2: We agree with this comment. Throughout the text, we exclusively used the term Weizmannia coagulans and noted in a single sentence that its former name was Bacillus coagulans (Lines 69-71).

Comments 3: The aim of research was to investigate the influence of W. coagulans PR06 fermentation on the texture characteristics, cooking quality and starch digestibility of Oolong tea fortified rice noodles. The aim is clear pointed out in line from 80 to 82. However, my suggestion is that author write aim as particular last paragraph of chapter Introduction.

Response 3: We appreciate your valuable suggestion. We have relocated the previous Lines 80-82 to the end of the last paragraph in the Introduction chapter, now reflected as Lines 81-84.

Reviewer 2 Report

Comments and Suggestions for Authors

The paper describes the effect of fermentation of the flour blend for rice noodle-making by W. coagulans on the quality characteristics of the Oolong tea-fortified rice noodles. Besides noodle quality attributes (cooking features, texture, antioxidant capacity), detailed analyses of changes in rice flour gel rheology, starch structure, and digestibility were performed. The experiment is well-designed, and the results are well-presented and discussed. However, some corrections are required.

Lines 272-273 The yellow color of a matrix in the CLSM image. Include a more detailed explanation and references regarding interpreting yellow colour as a complexed starch-protein matrix. It might be better to reference the work Jia et al. 2020 https://doi.org/10.1016/j.ijbiomac.2020.02.101 but other references and explanations are welcome.

In Lines 295-298, it is stated that all texture features of the studied noodles statistically significantly increased after fermentation. However, this statement is not entirely in line with the data presented in Table 1. From Table 1, it can be concluded that the majority of textural indices are not significantly different except springiness and chewiness at 3 and 5% IA. The mean values indeed tend to increase with fermentation but without reaching statistical significance. Please, check whether the significance is well presented in Table 1 or reformulate the statements regarding the effect of fermentation on noodle textural attributes.

It would be informative to include adhesiveness as a textural attribute.

I may note that TPA is not always the best choise for the evaluation of product textural parameters. I recommend that in future work other texture-measuring tests are considered that are specifically developed for estimation of pasta/noodle hardness and adhesiveness.

Lines 330-332: Please, give a more detailed explanation of the mechanism behind the anticipated influence of fermentation on the reaction between OH groups in tea polyphenols and starch molecules. Also, if possible include references.

Author Response

We would like to express our gratitude for your professional review, constructive comments, and valuable suggestions regarding our manuscript. We have systematically addressed and revised the manuscript based on your feedback. We look forward to your further review of our updated submission.

Comments 1: Lines 272-273 The yellow color of a matrix in the CLSM image. Include a more detailed explanation and references regarding interpreting yellow colour as a complexed starch-protein matrix. It might be better to reference the work Jia et al. 2020 https://doi.org/10.1016/j.ijbiomac.2020.02.101 but other references and explanations are welcome.

Response 1: We agree with the comment. Accordingly, the reference has been incorporated into the manuscript (Reference 26).

Comments 2: In Lines 295-298, it is stated that all texture features of the studied noodles statistically significantly increased after fermentation. However, this statement is not entirely in line with the data presented in Table 1. From Table 1, it can be concluded that the majority of textural indices are not significantly different except springiness and chewiness at 3 and 5% IA. The mean values indeed tend to increase with fermentation but without reaching statistical significance. Please, check whether the significance is well presented in Table 1 or reformulate the statements regarding the effect of fermentation on noodle textural attributes.

Response 2: Thank you for bringing this to our attention. We have revised the statements concerning the impact of fermentation on the textural attributes of noodles (Lines 299 and 303).

Comments 3: It would be informative to include adhesiveness as a textural attribute. I may note that TPA is not always the best choise for the evaluation of product textural parameters. I recommend that in future work other texture-measuring tests are considered that are specifically developed for estimation of pasta/noodle hardness and adhesiveness;

Response 3: Thank you for your valuable suggestion. In future work, we will conduct additional texture-measuring tests specifically designed to estimate the hardness and adhesiveness of pasta and noodle.

Comments 4: Lines 330-332: Please, give a more detailed explanation of the mechanism behind the anticipated influence of fermentation on the reaction between OH groups in tea polyphenols and starch molecules. Also, if possible include references.

Response 4: We greatly appreciate your valuable suggestion. To clarify the mechanism you mentioned, 1H-NMR measurements are necessary to determine the coupling constants J(H,H) in both non-fermented and fermented samples, to investigate the hydrogen bonding interactions between tea polyphenols and starch molecules. In future research, we plan to conduct additional experiments to explore the mechanism behind the expected influence of fermentation on the interaction between OH groups in tea polyphenols and starch molecules.

In addition, we have added the missing information for Reference 15.